# Dualities between 2+1d fusion surface models from braided fusion categories

Luisa Eck

Rudolf Peierls Centre for Theoretical Physics, Parks Rd, Oxford OX1 3PU, United Kingdom

January 27, 2025

## Abstract

Fusion surface models generalize the concept of anyon chains to 2+1 dimensions, utilizing fusion 2-categories as their input. We investigate bond-algebraic dualities in these systems and show that distinct module tensor categories $\mathcal{M}$ over the same braided fusion category $\mathcal{B}$ give rise to dual lattice models. This extends the 1+1d result that dualities in anyon chains are classified by module categories over fusion categories. We analyze two concrete examples: (i) a $\mathrm{Rep}(S_3)$ model with a constrained Hilbert space, dual to the spin-$\frac{1}{2}$ XXZ model on the honeycomb lattice, and (ii) a bilayer Kitaev honeycomb model, dual to a spin-$\frac{1}{2}$ model with XXZ and Ising interactions. Unlike regular $\mathcal{M} = \mathcal{B}$ fusion surface models, which conserve only 1-form symmetries, models constructed from $\mathcal{M} \neq \mathcal{B}$ can exhibit both 1-form and 0-form symmetries, including non-invertible ones.

## 1 Introduction

Dualities are ubiquitous in statistical mechanics and condensed matter physics, starting with the seminal 1941 work of Kramers and Wannier [1]. They identified a "symmetry property" that relates low- and high-temperature phases of the two-dimensional classical Ising model and used it to locate the critical temperature. The Kramers-Wannier duality is a prime example of a *non-invertible* mapping, which becomes a symmetry at the self-dual critical point of the Ising model. Over the past years, research on non-invertible symmetries has expanded significantly [2–6], beginning with the "topological symmetry" in the Fibonacci golden chain [7].

The exploration of duality mappings beyond Kramers-Wannier likely began with Temperley and Lieb's seminal 1971 paper [8]. They introduced what are now called Temperley-Lieb algebras, a family of algebras with a complex parameter, to relate the transfer matrix spectra of distinct statistical mechanical models. Baxter et al. [9] proposed an alternative graphical method to demonstrate such an "equivalence" between the $q$-state Potts model and the six-vertex model with a specific anisotropy parameter. As this example highlights, dualities can connect systems with different Hilbert spaces. Their energy levels ought to be the same under appropriate boundary conditions, although the degeneracies in the spectrum may be different. Bond-algebraic dualities have emerged as a unifying language for describing such relationships [10]. Dual Hamiltonians are decomposed into local terms that generate the same operator algebra, referred to as the *bond algebra*.

For 1+1d *anyon chains*, dual models are systematically constructed by choosing different module categories over the same fusion category [11, 12], which determines the bond algebra. This approach also provides direct access to a matrix product operator that implements the mapping between the

dual chain. The same mathematical framework also underpins a generalized Landau paradigm for classifying gapped phases [5, 13, 14]. In certain cases, duality mappings between 2d statistical mechanical models – which, in a specific limit, correspond to quantum anyon chains – can be formulated as lattice orbifolds [15, 16]. In the critical regime, these lattice constructions correspond to CFT orbifolds.

Extending this understanding to 2+1d lattice models remains an open challenge, though significant progress has been made, often framed in the language of gauge theories. Gauging global invertible symmetries in the context of projected entangled-pair states (PEPS), rather than Hamiltonians, has been explored in [17]. Barkeshli et al. [18] used $G$-crossed braided tensor categories to relate symmetry-enriched topological orders via gauging. Generalized transverse-field Ising models in 2+1d were gauged in [19] to construct systems with fusion 2-categorical symmetries.

In this paper, we explore dualities in 2+1d *fusion surface models* constructed from braided fusion categories, as introduced by Inamura and Ohmori [20]. In contrast to earlier approaches [17–19], which begin with systems exhibiting global invertible 0-form symmetries, our method begins with systems possessing categorical 1-form symmetries. The mathematical foundation for dualities in these models is provided by module tensor categories over braided fusion categories, which can be seen as a categorification of the module categories over fusion categories underpinning the 1+1d framework [11, 21]. Module tensor categories have also been utilized in [22, 23] to construct enriched string-net models, which share the same Walker-Wang bulk.

From a physical perspective, our approach is motivated by the search for tractable lattice models with rich phase diagrams that encompass topologically ordered, symmetry-broken, and gapless phases. In our earlier work [24], we explored fusion surface models built from braided fusion categories $\mathcal{B}$, which can be viewed as generalizations of Kitaev's honeycomb model and feature categorical 1-form symmetries $\mathcal{B}$. The examples we studied consistently exhibited a schematic phase diagram of the following form:

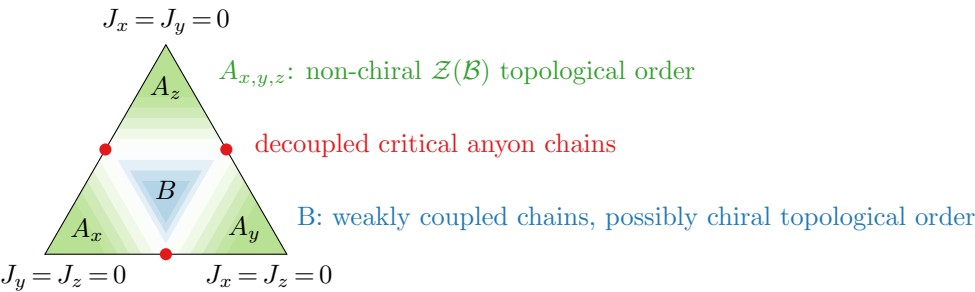

The triangular structure of this diagram with coupling constants $J_x + J_y + J_z = 1$ reflects the honeycomb geometry. The anisotropic phases $A_{x,y,z}$ display non-chiral $\mathcal{Z}(\mathcal{B})$ topological order when the local Hamiltonian is tuned to a projector onto the identity object. The isotropic phase $B$ is likely to realize chiral topological order when time-reversal is explicitly broken. For Kitaev's solvable honeycomb model, which can be formulated as an Ising fusion surface model [20], the phase boundaries are known exactly. For other examples, such as $\mathbb{Z}_3$ and Fibonacci generalizations of Kitaev's honeycomb model, numerical simulations and coupled wire arguments are needed to map out the phase diagram [24]. Although the existence of additional intermediate phases cannot be ruled out, none have been identified in our examples so far.

The framework developed here expands the scope of fusion surface models built from braided fusion categories by incorporating module categories, which permits the realization of models with 0-form symmetries and symmetry-enriched topological orders. Among the 2+1d models that fit into our framework are well-known systems like the spin-$\frac{1}{2}$ XXZ model on honeycomb or square

lattices. The 2+1d XXZ model is a paradigmatic example of a quantum spin system with anisotropic interactions, capturing rich phenomena such as spin-liquid behavior and symmetry-breaking phases [25–30]. Another notable example is a spin-$\frac{1}{2}$ model with $\mathbb{Z}_2$ 0-form and 1-form symmetries that is dual to a bilayer Kitaev honeycomb model. The bilayer Kitaev honeycomb model extends the celebrated Kitaev model to include interlayer couplings, offering a fertile ground to explore topologically ordered phases and anyon condensation transitions between them [31]. Its dual model exhibits a rich interplay of symmetry-broken and topologically ordered phases.

We begin by reviewing dualities in 1+1d anyon chains, using $\mathrm{Rep}(S_3)$ anyon chains as a guiding example. We then state our main result, an extension of this framework to 2+1d fusion surface models. Next, we explore two novel examples: the XXZ honeycomb model, dual to a constrained Hilbert space $\mathrm{Rep}(S_3)$ model, and a bilayer Kitaev honeycomb model, dual to an XXZ-Ising model. Finally, we analyze the symmetry fusion 2-category of the dual models from a mathematical perspective and conclude with a discussion of potential future directions.

## 2    Review: Dual 1+1d anyon chains

To prepare for the construction of dual 2+1d fusion surface models in the subsequent sections, we begin with a review of dualities in 1+1d anyon chains, following [11, 12]. For a more detailed introduction to the underlying mathematical framework – module categories over fusion categories – see, for instance, Chapter 7 of [32]. As a concrete example, we consider the $\mathcal{C} = so(3)_2/\mathrm{Rep}(S_3)$ anyon chain, which describes Rydberg-blockade atoms on a ladder, and its dual counterpart, the well-known spin-$\frac{1}{2}$ XXZ chain [11, 16].

**Hilbert space**    The construction of anyon chains, as described in [2, 4, 7] and reviewed in Section II of our earlier work [24], starts with two key pieces of input data: a fusion category $\mathcal{C}$ and a fixed object $\rho \in \mathcal{C}$. In this paper, we focus on unitary, multiplicity-free fusion categories with self-dual objects and trivial Frobenius-Schur indicators. The states in the Hilbert space are represented by fusion trees,

$$
|\{\Gamma_i\}\rangle = \quad
\begin{array}{c}
\Gamma_1 \quad \Gamma_2 \quad \Gamma_3 \quad \dots \quad \in \mathcal{C} \\
\rho \quad\quad \rho \quad\quad \rho \quad \in \mathcal{C}
\end{array}
\tag{1}
$$

In this diagram, the vertical legs are labeled by $\rho \in \mathcal{C}$, while the dynamical degrees of freedom, $\Gamma_i \in \mathcal{C}$, live on the horizontal dashed edges.

While fusion categories provide the necessary mathematical structure to define anyon chains with categorical symmetries, *module categories* over a given fusion category are essential to establish dualities between different anyon chains. As we demonstrate later in this section, the bond algebra of a Hamiltonian depends only on the fusion category, not on the module category, enabling the natural emergence of bond-algebraic dualities – such as the Temperley-Lieb ones discussed in the introduction – complete with explicit mappings between Hamiltonians. Certain dualities, like those between the Rydberg-blockade ladder and the $\mathcal{M} = \mathrm{Rep}(\mathbb{Z}_3)$ model (a 3-state Potts model with a hard antiferromagnetic constraint) can be understood in the framework of lattice orbifolds [15, 16]. In the continuum limit, these lattice orbifolds become orbifold transformations between conformal field theories. The advantage of the lattice orbifold approach lies in its simplicity: it does not require knowledge of module categories and can instead be visualized through geometric transformations of incidence graphs. However, this framework is not easily generalized to higher dimensions, which motivates our use of the more systematic approach based on module categories.

The input to the construction is refined by specifying a *module category* $\mathcal{M}$ over the fusion category $\mathcal{C}$. A right module category $\mathcal{M}$ consists of a set of objects $\{M, N, \dots\} \in \mathcal{M}$ equipped with a right action $M \lhd \alpha = \oplus_N N$ for objects $\{\alpha, \beta, \dots\} \in \mathcal{C}$. Pictorially, this action is represented by a trivalent vertex where two lines labeled by objects in the module category $\mathcal{M}$ meet a single line labeled by an object in the fusion category $\mathcal{C}$. The vector space $\mathcal{V}_{\alpha N}^M$ associated with such a vertex is typically multidimensional, even when the fusion category $\mathcal{C}$ is multiplicity-free. In the anyon chain constructed from the category pair $(\mathcal{M}, \mathcal{C})$, the degrees of freedom are now objects $M_i \in \mathcal{M}$ labeling the horizontal edges and basis vectors $v_{i,j} \in \mathcal{V}_{\rho M_j}^{M_i}$ living on the trivalent vertices,

$$|\{M_i\}, \{v_i\}\rangle = \qquad\qquad \begin{array}{c} M_1 \;\; v_{1,2} \;\; M_2 \;\; v_{2,3} \;\; M_3 \;\; v_{3,4} \;\; \dots \qquad \in \mathcal{M} \\[4pt] \\ \rho \qquad\quad \rho \qquad\quad \rho \qquad\qquad \in \mathcal{C} \end{array} \tag{2}$$

A key example of this extended anyon chain construction featuring module categories, analyzed in [11, 16, 21], is based on the fusion category $\mathcal{C} = \mathrm{Rep}(S_3)$ or equivalently $so(3)_2$. Its simple objects are $\{0, 1, 2\}$ with fusion rules

$$1 \otimes 1 = 0 \oplus 1 \oplus 2, \quad 2 \otimes 1 = 1, \quad 2 \otimes 2 = 0.$$

We follow the notation from [16], where the non-abelian "spin-1" object in $so(3)_2$ is denoted by 1. This differs from the convention in [11], where 2 represents the non-abelian object corresponding to the 2D irreducible representation of $S_3$. There are four module categories over $\mathcal{C} = \mathrm{Rep}(S_3)$, namely $\mathcal{M} = \mathrm{Vec}$, $\mathcal{M} = \mathrm{Rep}(\mathbb{Z}_2)$, $\mathcal{M} = \mathrm{Rep}(\mathbb{Z}_3)$, and $\mathcal{M} = \mathrm{Rep}(S_3)$. Each module category gives rise to a distinct anyon chain.

For instance, the regular module category $\mathcal{M} = \mathrm{Rep}(S_3)$ leads to an anyon chain with a constrained Hilbert space, which can be interpreted in terms of the Rydberg blockade mechanism observed in arrays of nearby excited Rydberg atoms. In this interpretation, the non-abelian object 1 represents an empty rung of a square ladder, while the abelian objects 0 and 2 correspond to excited atoms on the top and bottom sites of the rung, respectively [16, 33].

$$|1, 0, 1, 2, 1, \dots\rangle = \qquad \begin{array}{c} 1 \quad\; 0 \quad\; 1 \quad\; 2 \quad\; 1 \;\; \dots \quad \in \mathrm{Rep}(S_3) \\[4pt] \\ 1 \quad\; 1 \quad\; 1 \quad\; 1 \quad\; 1 \qquad \in \mathrm{Rep}(S_3) \end{array}$$

In contrast, the trivial module category $\mathcal{M} = \mathrm{Vec}$ yields an anyon chain with a tensor product Hilbert space of qubits $v_{i,i+1} \in \mathbb{Z}_2$ living on the trivalent vertices. All the horizontal edges are labeled by the unique object in $\underline{0} \in \mathrm{Vec}$, and therefore do not contribute dynamic degrees of freedom.

$$|\{v_i\}\rangle = \qquad \begin{array}{c} v_{1,2} \quad v_{2,3} \quad v_{3,4} \qquad \in \mathrm{Vec} \\[4pt] \\ 1 \qquad 1 \qquad 1 \qquad \in \mathrm{Rep}(S_3) \end{array} \qquad \text{with } v_{i,i+1} \in \mathbb{Z}_2. \tag{3}$$

This can be seen as follows: For any subgroup $A \subseteq S_3$, $\mathrm{Rep}(A)$ serves as a module category over $\mathrm{Rep}(S_3)$ via the restriction functor $\mathrm{Res}_A^{S_3} : \mathrm{Rep}(S_3) \to \mathrm{Rep}(A)$. The module action is defined as $M \lhd \alpha \equiv M \otimes \mathrm{Res}_A^{S_3}(\alpha)$ for all $M \in \mathrm{Rep}(A)$ and $\alpha \in \mathrm{Rep}(S_3)$ [32]. In our example of $\mathcal{M} = \mathrm{Vec} \simeq \mathrm{Rep}(\mathbb{Z}_1)$,

the restriction functor acts as $\mathrm{Res}^{S_3}_{\mathbb{Z}_1}(0) = \mathrm{Res}^{S_3}_{\mathbb{Z}_1}(2) = \underline{0}$ and $\mathrm{Res}^{S_3}_{\mathbb{Z}_1}(1) = 2 \cdot \underline{0}$ for $0, 1, 2 \in \mathrm{Rep}(S_3)$ and $\underline{0} \in \mathrm{Vec}$. The notation $2 \cdot \underline{0}$ means two copies of the object $\underline{0}$. In particular, this implies $\underline{0} \triangleleft 1 = 2 \cdot \underline{0}$, which results in the $\mathbb{Z}_2$ degrees of freedom $v_{i,i+1} \in \mathcal{V}^0_{1\underline{0}}$ on the trivalent vertices in (3).

**Hamiltonian and symmetries**  The action of the local Hamiltonian $H^{(\lambda)}_{i-1,i,i+1}$ with $\lambda \in \mathcal{C}$ on the general state (2) can be illustrated as

$$H^{(\lambda)}_{i-1,i,i+1}: \qquad \rightarrow \qquad . \tag{4}$$

Changing the module category $\mathcal{M}$ does not affect the *bond algebra* generated by the local terms $H^{(\lambda)}_{i-1,i,i+1}$, as it depends solely on the fusion category $\mathcal{C}$. The total Hamiltonian is given by

$$H = \sum_i c_i \sum_{\lambda \in \mathcal{C}} A_\lambda H^{(\lambda)}_{i-1,i,i+1}, \tag{5}$$

where $c_i$ and $A_\lambda$ are real constants.

To compute the local Hamiltonian (4) explicitly, we need the module category $^{\triangleleft}F$ symbols as defined in [11],

$$\qquad = \sum_C \sum_{i,l} [^{\triangleleft}F^{A\alpha\beta}_B]^{\gamma,k}_{C,i,l} \qquad . \tag{6}$$

Additionally, we assume completeness and orthogonality conditions for appropriately chosen basis vectors [34],

$$\qquad = \delta_{NM}\delta_{i,j}\sqrt{\frac{d_O d_\alpha}{d_M}} \quad , \qquad = \sum_{O,i} \sqrt{\frac{d_O}{d_M d_\alpha}} \qquad . \tag{7}$$

Using the identities (7) and (6), the local Hamiltonian (4) evaluates to

$$\qquad = \sum_{M'_i, k} \sqrt{\frac{d_{M'_i}}{d_{M_i} d_\lambda}}$$

$$= \sum_{\substack{M'_i, k \\ v'_{i-1,i}, v'_{i,i+1}}} [^{\triangleleft}F^{M_i \lambda \rho}_{M_{i-1}}]^{M'_i, v'_{i-1,i}, \bar{k}}_{\rho v_{i-1,i}} [^{\triangleleft}\bar{F}^{M_{i+1} \rho \lambda}_{M'_i}]^{M_i, v_{i,i+1}, k}_{\rho v'_{i,i+1}} \sqrt{d_\lambda}$$

The choice of a module category $\mathcal{M}$ also affects the symmetries of the system. For the regular module category $\mathcal{M} = \mathcal{C}$, the lattice model has a symmetry corresponding to each object in $\mathcal{C}$. Non-abelian objects give rise to non-invertible symmetries. For generic choices of $\mathcal{M}$, the symmetry category is no longer $\mathcal{C}$, but its Morita dual fusion category $\mathcal{C}^*_\mathcal{M} = \mathrm{End}_\mathcal{C}(\mathcal{M})$. The dual fusion categories $\mathcal{C}$ and $\mathcal{C}^*_\mathcal{M}$ share the same Drinfeld center, $\mathcal{Z}(\mathcal{C}) = \mathcal{Z}(\mathcal{C}^*_\mathcal{M})$. Graphically, the symmetry action is depicted as:

$$D^{(\alpha)}: \quad \overline{\underset{v_{i-1,i}}{M_{i-1}} \underset{}{M_i} \underset{v_{i,i+1}}{M_{i+1}}}^{\alpha} \quad \text{for } \alpha \in \mathcal{C}_\mathcal{M}^* \tag{8}$$

The Morita dual category $\mathcal{C}_\mathcal{M}^*$ is the unique fusion category that transforms $\mathcal{M}$ into an invertible $\mathcal{C}$–$\mathcal{C}_\mathcal{M}^*$ bimodule. The invertibility of the bimodule, combined with the pentagon equations, ensures that the Hamiltonian commutes with the symmetries. Invertible bimodules also characterize transparent gapped boundaries between 2+1d Levin-Wen string-net models, which allow for the tunneling of bulk excitations [35]. The diagram (8) can be evaluated explicitly by fusing the symmetry line to the horizontal edges $M_i \in \mathcal{M}$ and utilizing the bimodule F-symbols ${}^{\bowtie}F$,

$$\underset{B}{\overset{\alpha \ A \quad \beta}{\underset{C}{\diagup^k_l}}} = \sum_D \sum_{m,n} [{}^{\bowtie}F_B^{\alpha A \beta}]_{D,m,n}^{C,k,l} \ \underset{B}{\overset{\alpha \quad A \ \beta}{\underset{n}{\diagup^m D}}} \tag{9}$$

In the $\mathcal{C} = \mathrm{Rep}(S_3)$ example, the $\mathcal{M} = \mathrm{Vec}$ Hamiltonian is precisely the well-known spin-$\frac{1}{2}$ XXZ chain when choosing $A_0 = -A_2 = \Delta/\sqrt{2}$ and $A_1 = \sqrt{2}$ in (5). The Hamiltonian of the Rydberg-blockade ladder with the same constants $A_\lambda$ is written out explicitly in terms of hard-core bosonic operators in [16]. Both Hamiltonians can be decomposed into

$$H^{\mathrm{Rep}(S_3)} = \sum_j S_j + \Delta P_j, \text{ with } S_j = \sqrt{2} H^{(1)}_{j-1,j,j+1},$$

$$P_j = \frac{1}{\sqrt{2}} \left( H^{(0)}_{j-1,j,j+1} - H^{(2)}_{j-1,j,j+1} \right).$$

The $S_j$, $P_j$ operators generate the $so(3)_2$ BMW algebra enhanced by its Jones-Wenzl projector [16], both in their XXZ chain representation and in their Rydberg ladder representation. The Rydberg ladder exhibits a $\mathbb{Z}_2$ symmetry $D^{(2)}$ corresponding to $2 \in \mathrm{Rep}(S_3)$, along with a non-invertible symmetry $D^{(1)}$ corresponding to $1 \in \mathrm{Rep}(S_3)$, which obeys the same fusion algebra $D^{(1)} \times D^{(1)} = \mathbb{I} + D^{(1)} + D^{(2)}$ as the non-abelian object. In contrast, the XXZ chain features a $\mathcal{C}_\mathcal{M}^* = \mathrm{Vec}_{S_3}$ symmetry, which is Morita dual to the $\mathcal{C} = \mathrm{Rep}(S_3)$ symmetry of the Rydberg-blockade ladder.

**Duality transformations** Duality transformations between Hamiltonians constructed from different module categories $\mathcal{M}$ and $\mathcal{N}$ can be explicitly expressed as matrix product operators (MPOs) using the data of module functors $X \in \mathrm{Func}(\mathcal{M}, \mathcal{N})$ [11]:

$$O^{(X)} H^\mathcal{M} = H^\mathcal{N} O^{(X)}$$

In cases where $\mathcal{M} = \mathcal{C}$, as in our Rydberg ladder example, the MPO can be visualized similarly as a symmetry operator, with its entries determined solely by the ${}^{\triangleleft}F$ symbols:

$$O^{(X)}: \quad \overset{X \in \mathcal{M}}{\underset{\rho \quad \rho}{\overset{\Gamma_{i-1} \quad \Gamma_i \quad \Gamma_{i+1}}{\mid \quad \mid}}} = \sum_{\{M_i\}} \prod_i [{}^{\triangleleft}F_\rho^{M_i X \Gamma_{i+1}}]_{M_{i+1} v_{i+1}}^{\Gamma_i v_i} \ \overset{M_{i-1} \ M_i \ M_{i+1}}{\underset{v_{i-1,i} \quad v_{i,i+1}}{\mid \quad \mid}} \overset{\ }{\underset{\rho \quad \rho}{}}$$

The ability to express the duality transformation as an MPO is not inherently guaranteed by the bond algebra framework but is a direct advantage of the categorical construction.

**Phase diagrams**  The energy spectra of dual models contain the same energy levels, although with different degeneracies and possibly twisted boundary conditions in different symmetry sectors. Explicit mappings between the dual lattice models built from the $\mathcal{C} = so(3)_3/\mathrm{Rep}(S_3)$ category are discussed in detail in [12, 16]. Under the duality, the energy gap in the thermodynamic limit – and thus the notion of gapped versus gapless phases – ought to be preserved. When they are gapped, the dual models can break different symmetries though, and when they are gapless, they can realize different conformal field theories. For example, the XXZ chain and the Rydberg-blockade ladder are both critical in the regime $|\Delta| \leq 1$, but the Rydberg ladder is described by the orbifold of the free boson conformal field theory that characterizes the critical XXZ chain [16, 36, 37]. In the gapped $\Delta > 1$ regime, the XXZ chain has two antiferromagnetically ordered (AFM) ground states, whereas the Rydberg-blockade ladder has three ground states permuted by the non-invertible symmetry $D^{(1)}$, associated with the non-abelian object $1 \in \mathrm{Rep}(S_3)$. In the $\Delta < -1$ regime, both models possess exact ground states that maximize the $U(1)$ charge or, in the case of the Rydberg ladder, a non-invertible $U(1)$ remnant [16].

$$\tag{10}$$

# 3 Construction of dual 2+1d fusion surface models from braided fusion categories

This section presents our main result: a systematic method to construct 2+1d fusion surface models that are bond-algebraic duals of those derived from a given braided fusion category $\mathcal{B}$. Module tensor categories provide the necessary mathematical framework to define these dual fusion surface models. Compared to the 1+1d case, the 2+1d setting necessitates additional mixed $\tilde{F}$ symbols and braiding symbols between objects in $\mathcal{M}$ and $\mathcal{B}$ to compute the Hamiltonian explicitly. A more detailed mathematical discussion of module tensor categories, along with the symmetry fusion 2-category of the dual models, is deferred to the final section. A detailed analysis of the duality operators implementing the mapping, as well as the interplay of boundary conditions and symmetry sectors, is left for future work.

**Hilbert space**  Our starting point is the class of 2+1d fusion surface models constructed from a braided fusion category $\mathcal{B}$, as introduced by [20] and further explored in our previous work [24]. Analogous to the 1+1d fusion trees (1), the states in the Hilbert space of these 2+1d models correspond to the following honeycomb fusion diagrams:

$$|\{\Gamma_i, \Gamma_{ijk}\}\rangle = \qquad\qquad\qquad\qquad \tag{11}$$

The planar dotted edges are labeled by the dynamical degrees of freedom $\Gamma_i$ and $\Gamma_{ijk}$ taking values in $\mathbb{B}$. We use the notation $\Gamma_{ijk}$ to label specific edges that are surrounded by three edges $\Gamma_i$, $\Gamma_j$, and $\Gamma_k$. The distinction between $\Gamma_i$ and $\Gamma_{ijk}$ lies solely in their geometric configuration.

To construct dual 2+1d models, the key ingredients are *module tensor categories* $\mathcal{M}$ over the braided fusion category $\mathcal{B}$ [38]. These are module categories which possess their own intrinsic tensor structure. We define states in the Hilbert space of the dual models as fusion diagrams of the form:

$$|\{M_i, M_{ijk}, v_{ijk}\}\rangle = \quad \text{} \tag{12}$$

Here the planar red edges are labeled by the degrees of freedom $M_i$ and $M_{ijk}$ in $\mathcal{M}$. As before, vertical legs are labeled by a fixed object $\rho \in \mathcal{B}$ and gray vertices, where objects in $\mathcal{M}$ meet $\rho$, are assigned basis vectors $v_{ijk} \in \mathcal{V}_{\rho M_{ijk}}^{M_i}$. Notably, these vertices $v_{ijk}$ do not appear in the fusion surface model construction derived from multiplicity-free braided fusion categories $\mathcal{B}$ that we examined in our earlier work [24]; cf. (11). The primary distinction between the 1+1d and 2+1d settings is the presence of trivalent vertices where three objects in $\mathcal{M}$ meet, as illustrated in the diagram above. These vertices necessitate an intrinsic tensor product of the module category $\mathcal{M}$.

**Hamiltonian**  We use the same local Hamiltonian $H_p^{(\lambda)}$ as in [24], defined for a label $\lambda \in \mathcal{B}$ and a plaquette $p$. Its action on the state (12) is illustrated as

$$H_p^{(\lambda)}: \; -J_x \; \text{} \; - J_y \; \text{} \; - J_z \; \text{} \tag{13}$$

As in the 1+1d case, it is evident from the pictorial representation that the bond algebra of the local Hamiltonian terms $H_p^{(\lambda)}$ remains invariant under changing the module tensor category $\mathcal{M}$. The full Hamiltonian is given by

$$H = \sum_p C_p \sum_{\lambda \in \mathcal{B}} A_\lambda H_p^{(\lambda)} \quad \text{with } C_p, A_\lambda \in \mathbb{R}. \tag{14}$$

The z-link term corresponds directly to the local anyon chain Hamiltonian (4). For the x-link term, the Hamiltonian line $\lambda \in \mathcal{B}$ is fused to the lattice using the orthogonality relations (7),

$$\text{} = \sum_{B',C',D',F'} \sum_{k,l,m,n} \sqrt{\frac{d_{B'}d_{C'}d_{D'}d_{F'}}{d_B d_C d_D d_F d_h^4}} \quad \text{}. \tag{15}$$

To evaluate the diagram on the right hand side of (15), a new type of mixed F-symbol $\tilde{F}$ is required. This mixed $\tilde{F}$-symbol is well-defined only for module categories with an intrinsic tensor product structure,

$$\text{} = \sum_Q \sum_i [\tilde{F}_M^{N\alpha P}]_{Q,i}^{O,k} \quad \text{} \tag{16}$$

Finally, the half-braiding phases between objects in $\mathcal{M}$ and $\mathcal{B}$ are needed [38]:

$$\text{} = \Omega_{M,k}^{N\alpha} \quad \text{} \tag{17}$$

These mixed $\tilde{F}$-symbols have to satisfy consistency conditions that resemble the pentagon equation, involving $\tilde{F}$, $^{\triangleleft}F$ and $^{\triangleright}F$. We anticipate that the $\tilde{F}$-symbols can be explicitly determined by solving these consistency relations. Using the $^{\triangleleft}F$, $\tilde{F}$ and $\Omega$ symbols, the x-link term (15) evaluates to

$$\sum_{\substack{B',C'\\D',F'}} \sum_{\substack{i',j'\\k,l,m,n}} [^{\triangleleft}F_A^{\rho\lambda B'}]_{\rho,i'}^{B,k,i} \, [\tilde{F}_D^{B\lambda C'}]_{B',k}^{C,l} \, [\tilde{F}_{B'}^{C'\lambda D'}]_{C,l}^{D,m} [\tilde{F}_E^{D\lambda F'}]_{D',m}^{F,n} \, [^{\triangleleft}F_G^{F\lambda\rho}]_{F',j',n}^{\rho,j} \, \Omega_{C,\bar{l}}^{C\lambda} \sqrt{d_\lambda}$$

(18)

The y-link term can be computed very similarly to the x-link term. The regular $\mathcal{M}=\mathcal{B}$ fusion surface model has categorical 1-form symmetries corresponding to objects in $\mathcal{B}$ [20, 24]. The fusion 2-categorical symmetries of models with $\mathcal{M}\neq\mathcal{B}$ are discussed in the last section.

**Phase diagram** For the dual fusion surface models with Hilbert space given in Eq. (12), we expect a similar phase diagram. As we show below, the anisotropic limits $A_{x,y,z}$ are characterized by non-chiral $\mathcal{Z}(\mathcal{M})$ topological order. In phase $B$, the presence of additional 0-form symmetries, as discussed in the final section, allows for the possibility for topological order enriched by invertible or non-invertible 0-form symmetries.

In the extreme limit $J_x = J_y = 0$, the ground state of the honeycomb model (13) becomes the simultaneous ground state of all z-link Hamiltonians. We choose the z-link Hamiltonian to be the projector onto the identity object, $H^z = -J_z P^{(0)}$, which allows for straightforward identification of its ground state. This projector can be depicted as follows:

$$\sqrt{d_\rho}P^{(0)} : \quad \text{} \quad = \delta_{M,P}\,\delta_{i,j}\sum_{k,N'}\sqrt{\frac{d_{N'}d_N}{d_M^2}} \quad \text{}$$

(19)

Here we used the orthogonality and resolution of identity relations (7). Because of the two delta functions in (19), only those states with $M = P$ and $i = j$ have nonzero eigenvalue under the projector and qualify as ground states of $H^z = -J_z P^{(0)}$. The energy does not depend on the basis vector $i$ since the projector maps each state $|M,i,N,i,M\rangle$ to the superposition $\sum_{i',N'}\sqrt{d_{N'}}\,|M,i',N',i',M\rangle$. Hence, the superposition $\sum_{i',N'}\sqrt{d_{N'}}\,|M,i',N',i',M\rangle$ is the unique $+1$ ground state for a given $M$. Therefore, we expect one ground state for each element $M \in \mathcal{M}$, and the ground state subspace is a $\mathcal{M}$ string-net.

Because the perturbation theory Hamiltonian has to satisfy the same bond algebra as the regular $\mathcal{M} = \mathcal{B}$ perturbation theory Hamiltonian derived in [24], it has to act as

$$H_p^{\text{eff},\,\lambda} : \quad \text{} \quad \rightarrow \quad \text{}$$

(20)

The $\mathcal{B}$-loop acting on the $\mathcal{M}$ string-net can be resolved into $\mathcal{M}$-loops,

$$\text{}\,0_{\mathcal{M}} \quad = \quad \sum_{M\in 0_{\mathcal{M}}\otimes\lambda}\text{}\,M$$

Here $0_{\mathcal{M}}$ denotes the identity object of $\mathcal{M}$ and the equality follows from the orthogonality relations (7). Therefore, the effective model in perturbation theory is a $\mathcal{M}$ string-net with a commuting plaquette operator Hamiltonian, giving rise to $\mathcal{Z}(\mathcal{M})$ anyonic excitations.

# 4 Example: XXZ honeycomb model and $\mathrm{Rep}(S_3)$ fusion surface model

In this section, we examine two lattice models that share a common bond algebra determined by the input category $\mathcal{B} = \mathrm{Rep}(S_3)$. The first model, built from the trivial module category, corresponds to the well-known XXZ model on a honeycomb lattice. The second, derived from the regular module category, has a constrained Hilbert space and a $\mathrm{Rep}(S_3)$ 1-form symmetry. Since the duality ought to preserve the distinction between gapped and gapless phases, we can leverage known results about the XXZ honeycomb model to deduce properties of its dual $\mathrm{Rep}(S_3)$ model.

## 4.1 XXZ model from $\mathcal{M} = \mathbf{Vec}$

We begin by selecting the trivial module category $\mathcal{M} = \mathbf{Vec}$ over the fusion category $\mathcal{B} = \mathrm{Rep}(S_3)$ with symmetric braiding. The resulting fusion surface model has a tensor product Hilbert space of qubits $v_{ijk} \in \mathbb{Z}_2$ located at the trivalent vertices involving $\rho$. All planar edges are labeled by the unique object in Vec.

$$|\{v_{ijk}\}\rangle = \qquad \qquad \qquad \qquad \qquad \qquad \qquad \qquad \tag{21}$$

Here, the x-link and y-link terms in the Hamiltonian are identical to the z-link term, previously identified in [11] as the spin-$\frac{1}{2}$ XXZ Hamiltonian, when we choose the couplings $A_0 = -A_2 = \Delta/\sqrt{2}$ and $A_1 = \sqrt{2}$ in (13). Hence, the full Hamiltonian is the XXZ model on a honeycomb lattice,

$$
\begin{aligned}
H = J_x \sum_{i,j \in \text{x-link}} (X_i X_j + Y_i Y_j + \Delta Z_i Z_j) + J_y \sum_{i,j \in \text{y-link}} (X_i X_j + Y_i Y_j + \Delta Z_i Z_j) \\
+ J_z \sum_{i,j \in \text{z-link}} (X_i X_j + Y_i Y_j + \Delta Z_i Z_j)
\end{aligned}
\tag{22}
$$

This model exhibits a $U(1) \rtimes \mathbb{Z}_2$ 0-form symmetry generated by $\sum_j Z_j$ and $\prod_j X_j$, but no 1-form symmetries. This symmetry structure aligns with theoretical expectations: gauging the $\mathrm{Rep}(S_3)$ 1-form symmetry of the dual model described in the next subsection produces a $S_3$ 0-form symmetry, which is a finite subgroup of the full $U(1) \rtimes \mathbb{Z}_2$ 0-form symmetry.

Variants of the XXZ honeycomb model (22) have been extensively explored in the condensed matter literature, particularly in the isotropic case ($J_x = J_y = J_z$) and with additional nearest and next-nearest neighbor interactions stabilizing spin-liquid phases [27–30]. We note that the XXZ model on a square lattice can also be realized as a fusion surface model by omitting every other vertical leg in (21):

$$\sum_{\lambda \in \mathrm{Rep}(S_3)} A_\lambda \left( J_x \qquad \qquad + J_y \qquad \qquad \right)$$

At the isotropic point $J_x = J_y = J_z$, the 2d XXZ model with $\Delta > 0$ is known to possess antiferromagnetic order, both on the square lattice [26, 39, 40] and on the honeycomb lattice [25, 41]. The Néel-ordered ground states are aligned along the z-axis for $\Delta > 1$ and within the xy-plane for $0 < \Delta < 1$. In the latter case, the $U(1)$ symmetry is spontaneously broken, leading to gapless Goldstone modes. At the Heisenberg point $\Delta = 1$, the full $SU(2)$ symmetry is spontaneously broken, again resulting in gapless Goldstone excitations over the Néel-ordered ground states.

We now analyze the phase diagram of the XXZ honeycomb model (22) away from the isotropic point, where two analytically tractable limits shed light on the phase structure. First, in the anisotropic $J_z \gg J_x, J_y$ limit at $\Delta = 1$, the model reduces to a $\mathcal{Z}(\mathcal{M}) = \text{Vec}$ trivial string-net in perturbation theory (see (20)), resulting in a trivially gapped phase. This trivial phase is expected to persist for other values of $\Delta > -1$, provided the z-link local Hamiltonian retains a unique ground state. Second, when $J_x = 0$ and $J_y = J_z$, the model reduces to decoupled XXZ spin chains. These chains are critical and described by a free boson CFT in the regime $-1 < \Delta \leq 1$, cf. (10). The schematic phase diagram featuring these two limits and the isotropic phase with antiferromagnetic order is illustrated below:

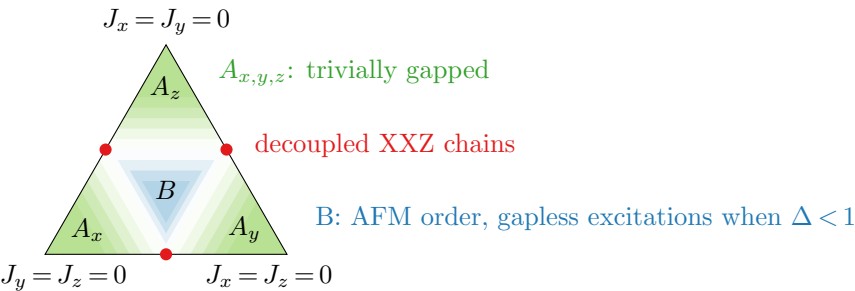

A second-order transition between the antiferromagnetic phase around the isotropic point and the trivial phase in the anisotropic limit has been observed numerically in the honeycomb XXZ model at $\Delta = 0$ [42]. We perform infinite DMRG simulations on the honeycomb Hamiltonian (22) with nonzero $\Delta$, fixing $J_y = J_z = 1$, and varying the rung coupling $J_x$. Calculations were performed using the TeNPy Library [43]. Our results are shown in Fig. 1 for $\Delta = 1$. To quantify quantum correlations both along the XXZ chain direction and around the short circumference of the cylinder, we measure the concurrence, as defined in [42].

$$C_{ij} = 2 \max \left\{ 0, |Z_{ij}| - \sqrt{X_{ij}^+ X_{ij}^-} \right\}, \quad \text{where } Z_{ij} = \langle S_{ij}^+ S_{ij}^- \rangle \text{ and } X_{ij}^\pm = \langle (1/2 - S_i^z)(1/2 \pm S_j^z) \rangle.$$

At large $J_x$, only the concurrence $C_x$ along the dimers of the unique ground state is nonzero. At smaller $J_x$, the concurrences in both directions are nonzero, as expected for antiferromagnetic ground states. The second derivative of the ground state energy peaks at the transition between these two regimes, signaling a critical phase transition. Around $J_x = 0$, zero concurrence along the rung hints that the system is effectively described by decoupled critical XXZ chains. However, no phase transition from this decoupled chains regime to the antiferromagnetic phase is observed in the energy derivatives, indicating that the decoupled chains regime does not represent a distinct quantum phase. In summary, our findings suggest that the qualitative phase diagram above and the second-order transition observed in [42] persist for nonzero $\Delta > 0$.

## 4.2 Fusion surface model from $\mathcal{M} = \text{Rep}(S_3)$

The regular $\mathcal{B} = \text{Rep}(S_3)$ fusion surface model features a constrained Hilbert space in which the fusion rules enforce $\Gamma_{ijk} \in \Gamma_i \otimes 1$ and $\Gamma_{ijk} \in \Gamma_j \otimes \Gamma_k$,

$$|\{\Gamma_i, \Gamma_{ijk}\}\rangle = \text{(diagram)} . \tag{23}$$

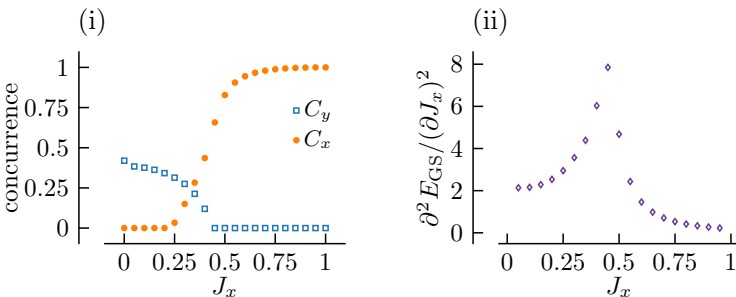

Figure 1: (i) Concurrences $C_x$, $C_y$ of the XXZ honeycomb model (22) and (ii) second derivative of the ground state energy, computed with infinite DMRG on a $L_y = 4$ cylinder with bond dimension $D = 800$. The plots are for fixed $J_y = J_z = (J_x - 1)/2$ and $\Delta = 1$.

The Hamiltonian (13) is invariant under a $\text{Rep}(S_3)$ 1-form symmetry and the corresponding mutually commuting plaquette operators, cf. (30). Time-reversal symmetry is preserved since all braiding phases and $F$-symbols are real. We choose the same parametrization as for the XXZ Hamiltonian, namely $A_0 = -A_2 = \Delta/\sqrt{2}$ and $A_1 = \sqrt{2}$ in (14).

This fusion surface model has the following two analytically tractable limits: In the anisotropic limit $J_z \gg J_x, J_y$ and $\Delta = 1$, the model simplifies to a $\text{Rep}(S_3)$ string-net exhibiting non-chiral $\mathcal{Z}(\text{Rep}(S_3))$ topological order [24]. The $\Delta = 1$ condition ensures that the z-link term acts as a projector matrix, which is used in the computation (20). As noted previously for the XXZ model, we expect this result to be robust around $\Delta = 1$, as long as the number of ground states of the local z-link Hamiltonian is the same. A similar $\mathcal{Z}(\text{Rep}(S_3))$ topologically ordered phase is expected when $J_x$ or $J_y$ dominate.

When $J_x = 0$ and $J_y = J_z$, the 2+1d model effectively reduces to a stack of $\text{Rep}(S_3)$ anyon chains summed over boundary conditions [24]. In the $-1 \leq \Delta \leq 1$ regime, this anyon chain, also known as the Rydberg-blockade ladder, is critical, as reviewed in (10). The phase $B$ in the center of the diagram is likely non-chiral, given that the Hamiltonian preserves time-reversal symmetry – unlike the chiral examples studied in [24]. Leveraging the duality to the XXZ model, we expect gapless excitations in this phase for $|\Delta| \leq 1$, akin to the Goldstone excitations seen in the XXZ model. We leave a further study of this phase for future work, as the constrained Hilbert space makes numerical simulations more challenging. The simplest phase diagram for fixed $\Delta > 0$ consistent with the above considerations is schematically

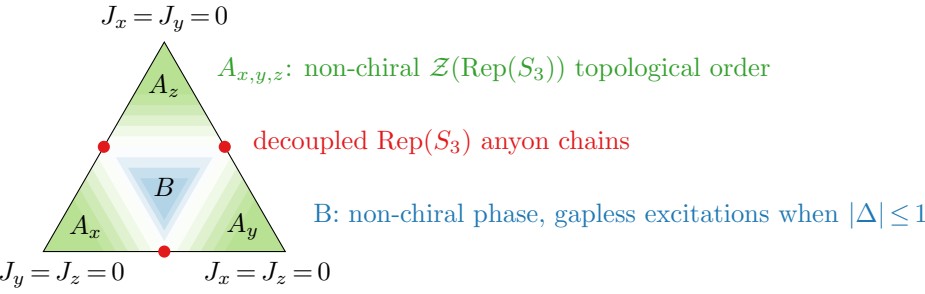

# 5 Example: Kitaev bilayer and XXZ-Ising honeycomb model

Next, we investigate a pair of models based on the $\mathcal{B} = \text{Ising} \boxtimes \overline{\text{Ising}}$ category. The regular module category gives rise to a bilayer Kitaev honeycomb model with $\mathbb{Z}_2 \boxtimes \overline{\mathbb{Z}_2}$ 1-form symmetries. Its dual counterpart, built from the $\mathcal{M} = \text{Ising}$ module category, resembles a honeycomb XXZ model augmented by additional link qubits with Ising interactions. This XXZ-Ising model preserves both $\mathbb{Z}_2$ 0-form and 1-form symmetries. Through perturbation theory and insights from the bilayer model, we explore the phase diagrams of these systems, identifying non-chiral topologically ordered phases and regions characterized by 0-form symmetry breaking.

## 5.1 Kitaev honeycomb bilayer model from $\mathcal{M} = \text{Ising} \boxtimes \overline{\text{Ising}}$

The input for constructing the fusion surface model is $\mathcal{B} = \text{Ising} \times \overline{\text{Ising}}$ and $\rho = \sigma\bar{\sigma}$, where $\sigma$ denotes the non-abelian object in the Ising fusion category, and the abelian objects are $\{0, 1\}$. The fusion rules are $\sigma \otimes \sigma = 0 \oplus 1$, $\sigma \otimes 1 = \sigma$, and $1 \otimes 1 = 0$. The objects in the $\overline{\text{Ising}}$ category with opposite braiding phases are denoted as $\bar{0}$, $\bar{1}$ and $\bar{\sigma}$.

For the regular module category $\mathcal{M} = \mathcal{B}$, the Hilbert space consists of two qubits on each orange dotted link $\Gamma_{ijk} \in \{0\bar{0}, 1\bar{0}, 0\bar{1}, 1\bar{1}\}$,

All blue links are fixed and labeled by the $\sigma\bar{\sigma}$ object. The Hamiltonian $H_p^{(\lambda)}$ for $\lambda = 1\bar{0}$ is unitarily equivalent to Kitaev's honeycomb model Hamiltonian [20, 24], acting only on the first layer of qubits. Similarly, $\lambda = 0\bar{1}$ acts on the second layer, while $\lambda = 1\bar{1}$ corresponds to the product of the two local Hamiltonians, as $0\bar{1} \otimes 1\bar{0} = 1\bar{1}$. Up to a unitary transformation, the total Hamiltonian is given by:

$$H = J_x \sum_{i,j\in\text{x-link}} \left( \sigma_i^x \sigma_j^x - \tau_i^x \tau_j^x + \Delta \sigma_i^x \sigma_j^x \tau_i^x \tau_j^x \right) + J_y \sum_{i,j\in\text{y-link}} \left( \sigma_i^y \sigma_j^y - \tau_i^y \tau_j^y + \Delta \sigma_i^y \sigma_j^y \tau_i^y \tau_j^y \right)$$
$$+ J_z \sum_{i,j\in\text{z-link}} \left( \sigma_i^z \sigma_j^z - \tau_i^z \tau_j^z + \Delta \sigma_i^z \sigma_j^z \tau_i^z \tau_j^z \right). \tag{24}$$

where $\sigma^\alpha$ and $\tau^\alpha$ act on the first and second qubit layer, respectively, and $\Delta$ is the interlayer coupling. The abelian objects in $\mathcal{B}$ give rise to a fermionic $\mathbb{Z}_2 \boxtimes \overline{\mathbb{Z}_2}$ 1-form symmetry, including commuting plaquette operators for each layer. The non-abelian objects do not generate 1-form symmetries, as they change the Hilbert space [24].

We chose the signs of the coupling constants in (24) to match the convention in Hwang [31], who numerically studied this Hamiltonian at isotropic couplings $J_x = J_y = J_z = 1$. Their results yield the following phase diagram:

Here we summarize their main findings: At $\Delta = 0$, the system consists of two decoupled gapless Kitaev honeycomb models. Introducing a small $\Delta$ induces chiral Ising topological order in both layers, but with opposite chiralities. Hence, this topologically ordered phase is described by the Ising $\boxtimes \overline{\text{Ising}}$ unitary modular tensor category, which served as input for our fusion surface model construction. This phase supports non-chiral, gapless edge modes characterized by the full Ising conformal field theory with central charge $c = (1/2, 1/2)$. In the large $\Delta$ limit, the system reduces, via perturbation theory, to a quantum dimer model on a kagome lattice with resonating valence bond ground states. This model is known to display toric code topological order $D(\mathbb{Z}_2)$ [44]. The second-order phase transition near $\Delta = 1$ can be interpreted as an anyon condensation transition between the Ising $\boxtimes \overline{\text{Ising}}$ and $D(\mathbb{Z}_2)$ topologically ordered phases, driven by the condensation of the bosonic $1\bar{1}$ anyon. The critical behavior at the transition point is believed to fall into the 3D Ising CFT universality class [45, 46].

Next, we discuss the phases of the bilayer model away from the isotropic point $J_x = J_y = J_y = 1$. In the regime where $J_z$ dominates over $J_x$ and $J_y$ with $\Delta > 0$, the system reduces to a bilayer toric code in perturbation theory, cf. (20). In this limit, the interlayer coupling $\Delta$ corresponds to the product of local terms from the two individual toric code Hamiltonians. A similar bilayer toric code model, albeit featuring a different Ising-like interlayer coupling, was analyzed in [47]. The authors identified a critical phase transition between a double toric code topological phase and a single toric code topological phase. An analogous transition is likely to occur in the anisotropic regime of our model when $\Delta$ is increased. When $J_x = 0$ and $J_y = J_z$, the bilayer model reduces to a stack of decoupled Ashkin-Teller chains, which are critical when $|\Delta| \leq 1$. Altogether, the triangular phase diagram of our bilayer Kitaev honeycomb model is expected to have the following structure for small $0 \leq \Delta \leq 1$,

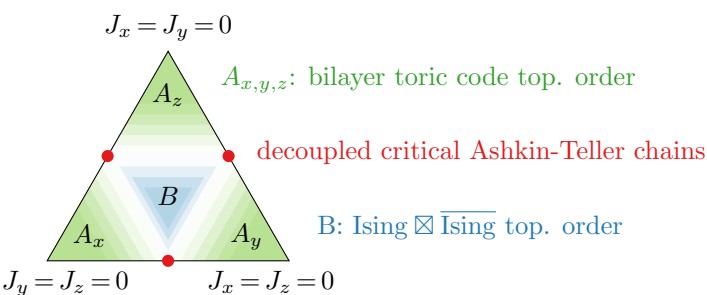

## 5.2 XXZ-Ising model from $\mathcal{M} = \text{Ising}$

Next, we select $\mathcal{M} = \text{Ising}$, which corresponds to the module category over the commutative algebra object $0\bar{0} \oplus 1\bar{1} \in \mathcal{B}$ [48]. The resulting Hilbert space consists of qubits $\Gamma_i \in \{0, 1\}$ on the trivalent vertices connected to $\rho$, and additional qubits $\Sigma_{ij} \in \{0, 1\}$ on the red links:

$$|\{\Gamma_i\}, \{\Sigma_{i,j}\}\rangle = \qquad\qquad\qquad\qquad\qquad\qquad (25)$$

The dashed black lines are labeled by $\sigma$, while the blue vertical legs are labeled by $\sigma\bar{\sigma}$.

The Hamiltonian is derived by resolving each $\sigma\bar{\sigma}$ leg into two separate $\sigma$ and $\bar{\sigma}$ legs and applying the standard F-symbols and R-symbols of the Ising category. For example, the z-link term with

$\lambda = 1\bar{0}$ can be computed as

$$H_{i,j}^{z,(1\bar{0})}: \quad  \quad = Y_i Z_j. \tag{26}$$

Following this procedure, the z-link terms corresponding to $\lambda = 0\bar{1}$ and $\lambda = 1\bar{1}$ can be derived similarly,

$$H_{i,j}^{z,(0\bar{1})} = Z_i Y_j, \quad H_{i,j}^{z,(1\bar{1})} = X_i X_j.$$

As before, the $1\bar{1}$-term is the product of the other two terms. With appropriate coupling constants and after a unitary rotation of the first qubit around the x-axis, the z-link Hamiltonian with all three terms is equal to the local XXZ Hamiltonian [11].

The x-link and y-link terms can be computed analogously. To render the Hamiltonian real, a unitary rotation $U$ is applied to all $\Gamma_i$ qubits on sublattice A and to all $\Sigma_{ij}$ qubits,

$$U = \prod_{i \in A} e^{i\pi\sigma_i^x/4} \prod_{ij} e^{i\pi\sigma_{ij}^z/4}. \tag{27}$$

After this rotation, the full honeycomb Hamiltonian takes the form (with the same choice of signs as in (24)):

$$
\begin{aligned}
H = J_z \sum_{\substack{i,j \\ \in \text{z-link}}} \left( Z_i Z_j - Y_i Y_j - \Delta X_i X_j \right) + J_x \sum_{\substack{i,ik,k \\ \in \text{x-link}}} \left( Y_i X_{ik} Y_k - Z_i X_{ik} Z_k - \Delta X_i X_k \right) \\
+ J_y \sum_{\substack{k,ik,lm,l \\ \in \text{y-link}}} \left( Y_k Z_{ik} Z_{lm} Y_l - Z_k Z_{ik} Z_{lm} Z_l - \Delta X_k X_l \right).
\end{aligned}
\tag{28}
$$

This Hamiltonian resembles an XXZ model on the honeycomb lattice, but with additional qubits placed on certain links, coupled by Ising interactions. It is real and therefore preserves time-reversal symmetry.

By resolving all $\sigma\bar{\sigma}$ legs into separate $\sigma$ and $\bar{\sigma}$ legs, as shown in (26), the model can also be interpreted as a regular $\mathcal{M} = \mathcal{B} = \text{Ising}$ fusion surface model, but defined on a different geometry and with longer-range interaction terms. Consequently, it preserves a $\mathbb{Z}_2$ 1-form symmetry, acting along the vertical incontractible loops as $\prod_{ij} X_{ij}$ and along the horizontal loops as $\prod_{ij} Z_{ij} \prod_i X_i$ (after the unitary rotation (27)). Since this 1-form symmetry is fermionic – and thus anomalous – it must be broken in all gapped phases, as discussed in [20].. In addition to the 0-form symmetries $\prod_{ij} X_{ij}$ and $\prod_i X_i$ generated by products of 1-form symmetry loops, we find an independent $\mathbb{Z}_2$ 0-form symmetry $\prod_i Z_i$ by inspection.

In the following, we provide arguments supporting the following phase diagram of the Hamiltonian (28):

In the $\Delta \to \infty$ limit, the vertex qubits form two ferromagnetic ground states $\langle X_i \rangle = \pm 1$ which break the $\prod_i Z_i$ 0-form symmetry, while the link qubits fluctuate freely. The lowest order effective Hamiltonian acting on this ground state subspace appears at sixth order and is equal to the conserved plaquette operator (analogous to the perturbation theory analysis in [31] for the dual bilayer model),

$$H_p^{(\text{eff})} \propto -\frac{J_x^2 J_y^2 J_z^2}{\Delta^5} \cdots \xrightarrow{\text{unitary rotation (27)}} Y_{im} Z_{kp} Y_{lq} Z_{jn} (X_i X_j X_k X_l).$$

Since the product over the four $X_i$ operators is always equal to $+1$ in the ferromagnetic ground states, this effective Hamiltonian is essentially the toric code Hamiltonian (in Wen's convention [49]) acting on the link qubits. Therefore, the large $\Delta$ phase has $D(\mathbb{Z}_2)$ topological order as well as a broken $\mathbb{Z}_2$ 0-form symmetry.

Characterizing the small $\Delta$ phase is more challenging, as it appears to resist analysis via standard perturbation theory, and further study would likely be needed. Based on the gauging analysis in [18], it is plausible that this phase corresponds to toric code topological order with unbroken 0-form symmetries. Their approach starts with a topological phase described by a UMTC $\mathcal{C}$ that also preserves an invertible 0-form symmetry $G$. Its symmetry-enriched class is described by a $G$-crossed braided tensor category $\mathcal{C}_G^X$, which incorporates both the anyons of $\mathcal{C}$ and extrinsic defects $\rho_{\mathbf{g}}$ associated with group elements $\mathbf{g} \in G$. These defects permute the anyons. Gauging $G$ leads to a new topological order $(\mathcal{C}_G^X)^G$, where the defects $\rho_{\mathbf{g}}$ become deconfined excitations. The data of $(\mathcal{C}_G^X)^G$ can be derived mathematically from $\mathcal{C}_G^X$, independent of specific microscopic realizations.

In our case, $(\mathcal{C}_G^X)^G = \text{Ising} \boxtimes \overline{\text{Ising}}$ characterizes the topological order of the bilayer Kitaev honeycomb model at small $\Delta$. This phase can emerge from gauging a $\mathcal{C} = D(\mathbb{Z}_2)$ toric code phase with a $G = \mathbb{Z}_2$ symmetry that permutes the bosonic $e$ and $m$ anyons (see Section I.2 in [18]). We therefore conjecture that this $D(\mathbb{Z}_2)$ phase enriched by the $\mathbb{Z}_2$ symmetry $G = \prod_i Z_i$ describes the XXZ-Ising model at small $\Delta$. The same phase has been realized in symmetry-enriched toric codes [50, 51]. With open boundary conditions, the XXZ-Ising model at small $\Delta$ must exhibit gapless non-chiral $c = (\frac{1}{2}, \frac{1}{2})$ Ising edge modes to match the gapless edge modes of the dual bilayer model.

The critical transition at $\Delta \approx 1$ therefore describes a ferromagnetic $\mathbb{Z}_2$ 0-form symmetry breaking transition, which corresponds to a 3d Ising CFT. Notably, the $D(\mathbb{Z}_2)$ topological order remains unchanged across the transition. This transition maps to an anyon condensation process in the dual Kitaev honeycomb bilayer model, also governed by a 3d Ising CFT. We provide numerical evidence for the ferromagnetic transition in Fig. 2, which shows the ferromagnetic order parameter alongside the second derivative of the ground-state energy. The DMRG algorithm spontaneously converges to one of the two ferromagnetic ground states, chosen at random. Consequently, we plot the expectation value $|\langle X_1 \rangle|$ at the first site rather than a connected two-point correlation function.

Finally, we turn to the triangular phase diagram of the XXZ-Ising model away from the isotropic point. When $J_z$ is dominant and $\Delta \geq 0$, vertex qubits on the same z-link are fixed to a unique dimer ground state $(|\uparrow\uparrow\rangle - |\downarrow\downarrow\rangle)/\sqrt{2}$, which minimizes the energy of their local Hamiltonian. The remaining degrees of freedom, represented by the link qubits, fluctuate, and the effective Hamiltonian obtained via perturbation theory corresponds to a toric code Hamiltonian acting on these link qubits.

While the anisotropic and isotropic regimes are conjectured to share the same $D(\mathbb{Z}_2)$ topological order (if the gauging analysis above holds for the isotropic phase), they should be distinct phases, separated by a phase transition akin to that of the dual bilayer model between the $\text{Ising} \boxtimes \overline{\text{Ising}}$ and bilayer toric code phases. In the anisotropic regime, the vertex qubits are frozen into dimers along the z-links, and only the link qubits contribute to the toric code topological order. This topological

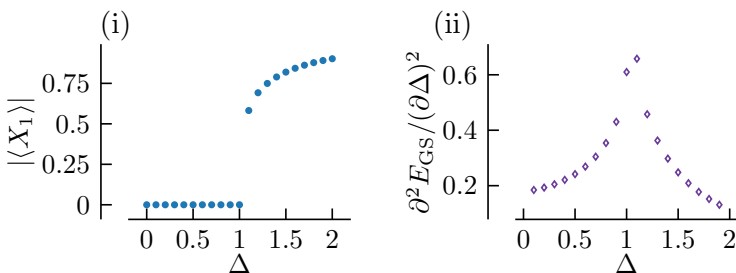

Figure 2: (i) Ferromagnetic order parameter $|\langle X_1 \rangle|$ of the XXZ-Ising model (28) and (ii) second derivative of the ground state energy, computed with infinite DMRG on a $L_y = 2$ cylinder with bond dimension $D = 200$. The plots are for fixed $J_y = J_z = J_x = 1$.

order can thus be characterized by the breaking of (emergent) 1-form symmetries restricted to the subsystem of link qubits. In contrast, near the isotropic point, the vertex qubits fluctuate freely, and the topological order arises from 1-form symmetry breaking across the entire system. Furthermore, gapless edge modes are expected near the isotropic point to match those of the bilayer Kitaev model, whereas the anisotropic regime should lack such edge modes. In summary, the simplest phase diagram for $0 \leq \Delta \leq 1$, consistent with the above analysis is given schematically as

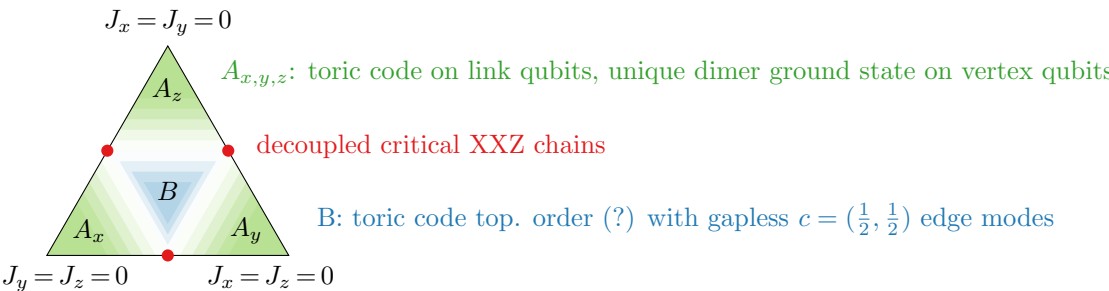

## 6   Symmetry fusion 2-category of dual models

In the following, we discuss the 0-form and 1-form symmetries of the dual fusion surface models, using a higher Morita theory for fusion 2-categories. Unlike regular fusion surface models with $\mathcal{M} = \mathcal{B}$, which preserve only 1-form symmetries and their associated condensation defects, dual models with $\mathcal{M} \neq \mathcal{B}$ support independent 0-form symmetries. Identifying the 0-form symmetries and their lattice action is more complex than determining the 1-form symmetries, and warrants further investigation [52, 53].

**Mathematical background**   Module tensor categories $\mathcal{M}$ over a braided fusion category $\mathcal{B}$ are also referred to as $\mathcal{B}$-enriched or $\mathcal{B}$-central tensor categories in the literature. For a detailed exposition, see [38, 54–57]. Following [38], a tensor category $\mathcal{M}$ becomes a module tensor category over a braided category $\mathcal{B}$ if it admits a braided functor

$$F^{\mathcal{Z}} : \mathcal{B} \rightarrow \mathcal{Z}(\mathcal{M}), \tag{29}$$

where $\mathcal{Z}(\mathcal{M})$ is the Drinfeld center of $\mathcal{M}$. This functor equips $\mathcal{M}$ with the structure of a $\mathcal{B}$-module category through the action $b \triangleleft m \equiv F(b) \otimes m$ for $b \in \mathcal{B}$, $m \in \mathcal{M}$. Here, $F$ is the composition of $F^{\mathcal{Z}}$

with the forgetful functor from $\mathcal{Z}(\mathcal{M})$ to $\mathcal{M}$. The functor $F^{\mathcal{Z}}$ also induces a half-braiding between objects in $\mathcal{B}$ and objects in $\mathcal{M}$.

An important class of examples are module tensor categories constructed from *commutative algebra objects* in $\mathcal{B}$ [38]. For a given braided category $\mathcal{B}$, the category $\mathcal{M} = \mathrm{Mod}_{\mathcal{B}}(A)$ of $A$-modules in $\mathcal{B}$ forms a module tensor category if $A \in \mathcal{B}$ is a commutative algebra object. When $A$ is not commutative, the category $\mathrm{Mod}_{\mathcal{B}}(A)$ is still a valid module category, but not a tensor category itself. The free module functor $F : \mathcal{B} \to \mathcal{M} : F(x) = A \otimes x$ serves as the braided central functor in this construction. Commutative algebra objects in multiplicity-free modular fusion categories of rank up to 9 have been classified in [48, 58–60]. In the physics literature, module tensor categories over braided fusion categories have been used to study enriched string-nets with chiral topological order [22, 23] and boundaries of 3+1d Walker-Wang models [61].

**Symmetries of the regular fusion surface model**  Mathematically, the symmetry fusion 2-category of the regular $\mathcal{M} = \mathcal{B}$ fusion surface model is the condensation completion $\mathrm{Mod}(\mathcal{B})$ of the braided fusion 1-category $\mathcal{B}$ [20]. The objects of $\mathrm{Mod}(\mathcal{B})$ are separable algebras $A \in \mathcal{B}$, and morphisms between two such algebras $A$ and $C$ form the category of $C$-$A$ bimodules, denoted ${}_C\mathcal{B}_A$ [62–65]. The fusion 2-category $\mathrm{Mod}(\mathcal{B})$ is connected, meaning there always exists a 1-morphism between any two objects. When $A$ and $C$ correspond to the identity object $0 \in \mathcal{B}$, the bimodule category ${}_0\mathcal{B}_0$ reduces to $\mathcal{B}$ itself.

Physically, this implies that the system's 1-form symmetries are labeled by the objects $\alpha \in \mathcal{B}$. These symmetries can act on incontractible loops $l$ or manifest as mutually commuting plaquette operators $B_p^{(\alpha)}$:

$$B_p^{(\alpha)} : \quad\quad\quad\quad\quad\quad\quad\quad , \quad W_l^{(\alpha)} : \quad\quad\quad\quad\quad\quad\quad\quad \tag{30}$$

The condensation defects $D^{(A)}$ are labeled by separable algebra objects $A \in \mathcal{B}$ [66–68],

$$D^{(A)} : \quad\quad\quad\quad\quad\quad\quad\quad\quad$$

The lattice action of both condensation defects and 1-form symmetries can be computed explicitly using the F-symbols and R-symbols of $\mathcal{B}$ [20, 24].

**Symmetries of fusion surface models with $\mathcal{M} \neq \mathcal{B}$**  Determining the symmetry fusion 2-category of fusion surface models with a different module tensor category $\mathcal{M} \neq \mathcal{B}$ becomes more intricate. However, the task remains feasible because braided fusion categories are known to be fully dualizable [55, 69]. By analogy with 1+1d systems, the symmetry of the dual lattice model is conjectured to correspond to the Morita dual of the fusion 2-category $\mathrm{Mod}(\mathcal{B})$ [11, 19, 52]. Formally, the Morita dual of $\mathrm{Mod}(\mathcal{B})$, relative to its module 2-category $\mathrm{Mod}(\mathcal{M})$, is defined as [70, 71]:

$$\mathrm{Mod}(\mathcal{B})^*_{\mathrm{Mod}(\mathcal{M})} = \mathrm{End}_{\mathrm{Mod}(\mathcal{B})}(\mathrm{Mod}(\mathcal{M})). \tag{31}$$

For 1-categories, the Morita dual $\mathcal{C}^*_{\mathcal{M}} = \mathrm{End}_{\mathcal{C}}(\mathrm{Mod}_{\mathcal{C}}(A))$ can be regarded as the category $\mathrm{Bimod}_{\mathcal{C}}(A)^{\mathrm{mop}}$ of $A$-$A$ bimodules in $\mathcal{C}$, with "mop" indicating the multiplication opposite. Analogously, the dual fusion 2-category (31) is equivalent to [55, 70, 71]

$$\mathrm{Mod}(\mathcal{B})^*_{\mathrm{Mod}(\mathcal{M})} \simeq \mathrm{Bimod}_{\mathrm{Mod}(\mathcal{B})}(\mathcal{M})^{\mathrm{mop}}. \tag{32}$$

The 2-category on the right-hand side represents the 2-category of $\mathcal{B}$-centered $\mathcal{M}$-$\mathcal{M}$-bimodule 1-categories [52]. This structure is discussed in Sections 3.5 and 3.6 of [55] and Section 3.4 of [72]. Its objects are $\mathcal{B}$-centered $\mathcal{M}$-$\mathcal{M}$-bimodules and 1-morphisms are functors between these bimodules. The term "$\mathcal{B}$-centered" implies the existence of an isomorphism $\eta : n \otimes b \simeq b \otimes n$ between objects $n$ in the $\mathcal{M}$-$\mathcal{M}$-bimodule and objects $b \in \mathcal{B}$, subject to certain coherence conditions.

The 1-form symmetries of the lattice model correspond to the bimodule endofunctors of the monoidal unit of $\mathrm{Bimod}_{\mathrm{Mod}(\mathcal{B})}(\mathcal{M})^{\mathrm{mop}}$. The monoidal unit is $\mathcal{M}$ itself, equipped with its canonical $\mathcal{B}$-centered $\mathcal{M}$-$\mathcal{M}$-bimodule structure. By Lemma 3.2.1 in [70], its endomorphism 1-category is equivalent to

$$\mathrm{Bimod}_{\mathrm{Mod}(\mathcal{B})}(\mathcal{M})^0 \simeq \mathrm{Mod}\left(\overline{\mathcal{Z}_{\mathcal{B}}(\mathcal{M})}\right), \tag{33}$$

where the overline denotes the braiding opposite. The fusion 1-category $\mathcal{Z}_{\mathcal{B}}(\mathcal{M})$ is the subcategory of $\mathcal{Z}(\mathcal{M})$ consisting of objects that braid trivially with the image of $\mathcal{B}$ under the functor $F^{\mathcal{Z}}$ defined in (29). Physically, this condition implies that the 1-form symmetries can be deformed freely across the lattice [52].

When $\mathcal{B}$ is non-degenerate, the Drinfeld center of $\mathcal{M}$ factorizes as [73]

$$\mathcal{Z}(\mathcal{M}) = \mathcal{B}^{\mathrm{mop}} \boxtimes \mathcal{Z}_{\mathcal{B}}(\mathcal{M}), \tag{34}$$

making the computation of $\mathcal{Z}_{\mathcal{B}}(\mathcal{M})$ straightforward.

If and only if the functor $\mathcal{F}^{\mathcal{Z}}$ is fully faithful, the symmetry fusion 2-category is connected and described by (33) (Corollaries 3.1.5 and 3.2.6 in [70]). In this case, there are no 0-form symmetries beyond the condensation defects associated with the 1-form symmetries in $\mathcal{Z}_{\mathcal{B}}(\mathcal{M})$. The fully faithfulness of $\mathcal{F}^{\mathcal{Z}}$ is assumed in the enriched string-net construction in [22, 23].

However, in general, $\mathcal{F}^{\mathcal{Z}}$ is not fully faithful. In such situations, the connected components of the symmetry fusion 2-category (32), corresponding to 0-form symmetries modulo condensation defects, remain incompletely understood and are only known in specific examples (cf. Remark 3.1.6 in [70]). Nonetheless, it is known that any fusion 2-category is Morita dual to a connected fusion 2-category $\mathrm{Mod}(\mathcal{B})$ (Theorem 4.2.2 in [70]). This means that any fusion 2-categorical symmetry can be realized as the symmetry of a $(\mathcal{M}, \mathcal{B})$ fusion surface model.

Next, we discuss some specific examples tied to the lattice models studied in the previous sections. For instance, when $\mathcal{B} = \mathrm{Rep}(G)$ and $\mathcal{M} = \mathrm{Vec}$, the dual fusion 2-category is

$$\mathrm{Bimod}_{\mathrm{Mod}(\mathcal{B})}(\mathcal{M}) \simeq 2\mathrm{Vect}_G, \tag{35}$$

which has $|G|$ connected components (Example 5.1.9 in [71]). The $\mathcal{B} = \mathrm{Rep}(S_3)$, $\mathcal{M} = \mathrm{Vec}$ fusion surface model – equivalent to the XXZ honeycomb model – is an example for this dual $G = S_3$ 0-form symmetry.

For the XXZ-Ising model built from $(\mathcal{B} = \mathrm{Ising} \boxtimes \overline{\mathrm{Ising}}, \mathcal{M} = \mathrm{Ising})$, the general analysis of the dual fusion 2-category suggests that no 0-form or 1-form symmetries are present: The functor $\mathcal{F}^{\mathcal{Z}} : \mathcal{B} \to \mathcal{Z}(\mathcal{M})$ is fully faithful because $\mathcal{B} \simeq \mathcal{Z}(\mathcal{M})$ in this case. Consequently, the symmetry fusion 2-category is connected and equivalent to $\mathrm{Mod}(\mathcal{Z}_{\mathcal{B}}(\mathcal{M}))$, which rules out independent 0-form symmetries. Since $\mathcal{B}$ is non-degenerate, the factorization of the Drinfeld center (34) implies $\mathcal{Z}_{\mathcal{B}}(\mathcal{M}) \simeq \mathrm{Vec}$, meaning that no 1-form symmetries arise. The apparent discrepancy between this categorical analysis and the observed 1-form and 0-form symmetries likely arises from the special nature of Ising fusion surface models. Specifically, certain planar edges in the Hilbert space are fixed to $\sigma$ instead of being treated as dynamical degrees of freedom.

# 7 Conclusions and outlook

We investigated bond-algebraic dualities in 2+1-dimensional quantum lattice models, starting from models with categorical 1-form symmetries. We demonstrated that the appropriate mathematical framework for classifying such dualities is provided by module tensor categories $\mathcal{M}$ over braided fusion categories $\mathcal{B}$. The dual fusion surface models constructed from the pair $(\mathcal{M}, \mathcal{B})$ are invariant under 0-form and 1-form symmetries, which form a fusion 2-category Morita dual to $\mathrm{Mod}(\mathcal{B})$ with respect to the module 2-category $\mathrm{Mod}(\mathcal{M})$.

To illustrate the general method, we presented four concrete lattice models grouped into two pairs, with each pair sharing a common bond algebra. First, we identified the spin-$\frac{1}{2}$ XXZ honeycomb model as the dual of a $\mathrm{Rep}(S_3)$ fusion surface model with a constrained Hilbert space. The original $\mathrm{Rep}(S_3)$ model exhibited categorical 1-form symmetries, while the dual XXZ model preserved an $S_3$ 0-form symmetry but no 1-form symmetries. In the second example, we studied a bilayer Kitaev honeycomb model with a $\mathbb{Z}_2 \boxtimes \overline{\mathbb{Z}_2}$ 1-form symmetry, which mapped to an XXZ-like model with additional Ising qubits. This dual model retained both $\mathbb{Z}_2$ 0-form and 1-form symmetries. We further analyzed the phase diagrams of these examples, drawing on known results from the literature, numerical simulations, and the gap-preserving nature of the duality mappings.

Several promising directions remain for future exploration: One immediate direction is the investigation of dual fusion surface models with non-invertible 0-form symmetries in addition to invertible or non-invertible 1-form symmetries. Such models could realize novel phases of topological order enriched by non-invertible 0-form symmetries, extending beyond the $G$-crossed braided tensor category framework for topological phases enriched by invertible symmetries [18]. A key challenge in this pursuit is the computation of the mixed F-symbols $\tilde{F}$ (16) from consistency conditions analogous to the pentagon equation. In 1+1 dimensions, certain dualities can be implemented using constant-depth unitary circuits with measurements [74] or sequential circuits [75]. A natural question is whether our 2+1-dimensional dualities can be implemented in a similar manner. Furthermore, 1+1d dualities have been shown to improve the efficiency of DMRG simulations by reducing entanglement growth [21]. Extending this method to the higher-dimensional setting could enhance numerical methods for studying 2+1d quantum lattice models. Finally, an intriguing phenomenon in both XXZ chains and their higher-dimensional generalizations on cubic lattices is the presence of quantum scars. These are associated with spin-helix exact eigenstate and have been studied theoretically and experimentally in [76]. A systematic investigation of quantum scar states in 2+1d lattice models like fusion surface models would be particularly interesting, as they exemplify weak ergodicity breaking and are experimentally accessible in cold-atom platforms [76].

**Acknowledgments** I would like to thank Paul Fendley for his helpful feedback on the manuscript and guidance throughout this work. I also thank Kansei Inamura for sharing valuable insights on fusion 2-categories. Additionally, I thank Kyle Kawagoe, Laurens Lootens, Sahand Seifnashri, Thomas Wasserman, André Henriques and Sakura Schafer-Nameki for inspiring discussions. This work has been supported in part by the EPSRC Grant no. EP/S020527/1, and in part by grant NSF PHY-2309135 to the Kavli Institute for Theoretical Physics (KITP).

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
