# Peer review of "Dualities between 2+1d fusion surface models from braided fusion categories"

_SciPost Physics_

## Round 2 · Referee Report · Anonymous (Referee 1) · 2025-8-4

Strengths

This paper is an interesting addition to the growing literature on lattice models with generalized symmetries. The paper describes an approach to take as input a model built form a symmetric braided fusion category and systematically produce gauged models by picking a choice of module tensor category M. This is expected given the results on anyon chains by Lootens-Delcamp-Verstraete and also on 2D lattice systems by Inamura-Ohmori and Delcamp-Tiwari.
The models studied constructed in this work seem interesting particularly owing to the possibility of chiral topological order.

Weaknesses

The analysis of the dual models from the symmetry perspective seems quite inadequate. I thought the main point was to understand the dual models as stabilized by certain generalized symmetries and relatedly understand the orders from a symmetry perspective. This has not been discussed much.

In fact, the full symmetry category even for the regular module category has not been discussed (see questions below).

Report

I would request the author to consider addressing the following questions before recommending the manuscript for publication.

Requested changes

1. Could the discussion in Sec. 6 be expanded by deriving the symmetry structure on some simple dual models? It would be very useful to connect the abstract math discsussion in the section to concrete lattice realization of operators.
2. The models built from B as input should have the full 2-category Mod B as defects. This includes all condensations. How are these implemented on the Inamura Ohmori model (with a choice of M being the regular module category)?
3. Similar to the above question, the B based Inamura Ohmori model with M as regular module category potentially hosts invertible G symmetries that involve various data (action on anyons, symmetry fractionalization etc.) as described by Barkeshli et al . How do we realize symmetry operators in the Inamura Ohmori model with non-trivial fractionalization class and relatedly how does one gauge such symmetries by changing M?
4. In Sec. 4.2, similar to the Kitaev model, could one deform phase B to a chiral topological order by a time reversal breaking perturbation. Is there some intuition about what this topological order might be?

Recommendation

Ask for minor revision

  • validity: good
  • significance: good
  • originality: ok
  • clarity: high
  • formatting: good
  • grammar: excellent

Author:  Luisa Eck  on 2025-10-10  [id 5914]

(in reply to Report 1 on 2025-08-04)
Disclosure of Generative AI use

The comment author discloses that the following generative AI tools have been used in the preparation of this comment:

I used ChatGPT 5 for language polishing and organizational edits; all analyses and conclusions are my own.

I thank the reviewer for the constructive feedback and insightful questions. Regarding the requested changes:

  1. Thank you for the suggestion. I have expanded the two brief examples at the end of Section 6 into a dedicated subsection with two more examples, one based on $\mathcal{B}=\text{Rep}(S_3)$ and $M=\text{Vec}(\mathbb{Z}_3)$ and another one with $\mathcal{B}=su(3)_1$ and $\mathcal{M}=TY(\mathbb{Z}_3)$. In general, determining the symmetry fusion 2-category from $\mathcal{M}$ and $\mathcal{B}$ is a challenging mathematical task and outside the scope of this paper; the aim here is to present concrete 2+1D models that fit into this extended fusion surface model framework and are related by dualities.
  2. The condensation defects correspond to hexagonal nets whose edges are labeled by the algebra object; this is now stated explicitly in Eq. (31) and the preceding paragraph. Their action is computed by fusing the net into the 2+1D fusion tree and evaluating with the F- and R-symbols.
  3. Taking the braided fusion category $\mathcal{B}$ and the regular module category as input to the fusion surface model construction does not guarantee global 0-form symmetries (beyond those realized as condensation defects of 1-form symmetries), though they can exist accidentally or emerge at low energies. To build 0-form symmetries in from the start, one could instead choose a different $\mathcal{B}$-module category or begin with a fusion 2-category that is not connected. In particular, $G$ symmetry-enriched string-nets can be regarded as fusion surface models constructed from a $G$-graded fusion category regarded as a separable algebra over the fusion 2-category 2Vec$_G$, as explained in Section 5.3 in Inamura & Ohmori. The symmetry fractionalization class will be determined by the input data in this case. Gauging these models amounts to specifying a module 2-category of the input fusion 2-category, which has been explored in more detail in https://arxiv.org/pdf/2506.09177. I have clarified the connection to symmetry-enriched topological order in the Conclusions.
  4. In Section 4.2 I take $\mathcal{B}=\text{Rep}(S_3)$ with the symmetric braiding to realize the duality to the XXZ model. Since all braiding phases are real, the Hamiltonian and any perturbations compatible with the fusion surface construction are real and time-reversal invariant, so I do not expect chiral topological order. I have added this clarification to Section 4.2. In contrast, if one uses $\mathcal{B}=\text{Rep}(S_3)$ with a non-degenerate braiding, the complex R-symbols generally lead to a fusion surface Hamiltonian that breaks time reversal (analogous to the $\mathbb{Z}_3$ Kitaev honeycomb model), making it a candidate for chiral topological order. Because the underlying $\text{Rep}(S_3)$ anyon chain is described by the c=1 free boson orbifold CFT at criticality, I would then expect chiral edge modes with $c=1$; a definite identification would require numerical simulations.

---

## Round 2 · Referee Report · Anonymous (Referee 2) · 2025-9-16

Strengths

1- Clear exposition of relevant background material, accessible to non-experts 2- Explicit examples to showcase the general theory, supplemented with numerical results 3- Allows the systematic construction of lattice models exhibiting chiral topological order

Weaknesses

1- In principle, the results in this work follow from the established general theory for dualities of models with fusion 2-categorical symmetries, although making these general statements concrete is a non-trivial exercise.

Report

This manuscript provides an explicit realisation of a class of (2+1)d lattice models and their dualities. These models are generalized Kitaev models obtained by choosing an input braided fusion category which dictates the bond algebra of local operators, together with a choice of module tensor category that provides the Hilbert space and an explicit realisation of this algebra of local operators. By keeping the bond algebra fixed and changing the module tensor category, we end up with a class of dual models that exhibit the same spectrum. This is an extension of the results on dualities (1+1)d anyonic spin chains described by fusion categories to (2+1)d quanum lattice models described by a certain class of fusion 2-categories.

In some sense, these results are a straightforward generalization of the (1+1)d story, and in this sense they come as expected. Nevertheless, I think it is still a valuable addition to the literature as the (expected) theoretical results are applied to explicit models, which are then explored using numerical methods. In particular, the capability of this framework for producing chiral topologically ordered lattice Hamiltonians provides an interesting playground for benchmarking numerical methods such as PEPS, for which chiral topological phases pose particular difficulties.

For these reasons, I think this paper is suitable for publication in Scipost Physics. Below, I list a few questions and comments that maybe be helpful in further improving the manuscript.

Requested changes

1- In (1+1)d, the duality operators are labelled by module functors between the relevant module categories that define the dual models, and as a quantum circuit they are linear depth, compatible with the fact that they change the phase of the states on which they act. Can one say anything about the depth of the duality operators in the (2+1)d setting? Similarly, the action of the duality operators on the phase of a model (which I suppose in (2+1)d is labelled by a module 2-category) can be understood via composotion of module functors, does the same hold here?

2- Using these module tensor categories, can one construct the internal Hom to obtain a higher dimensional generalisation of a special symmetric Frobenius algebra? In (1+1)d, these special symmetric Frobenius algebras provide the RG fixed point Hamiltonians associated to the phase determined by the module category used to construct the internal Hom, and one can obtain phase transitions by interpolating between these fixed point Hamiltonians. The (2+1)d version of this could shed some light on the possible phase transitions in these models.

Recommendation

Publish (meets expectations and criteria for this Journal)

  • validity: top
  • significance: good
  • originality: good
  • clarity: top
  • formatting: excellent
  • grammar: excellent

Author:  Luisa Eck  on 2025-10-10  [id 5915]

(in reply to Report 2 on 2025-09-16)
Disclosure of Generative AI use

The comment author discloses that the following generative AI tools have been used in the preparation of this comment:

I used ChatGPT 5 for language polishing and organizational edits only; all analyses and conclusions are my own.

I thank the reviewer for their thoughtful questions.

  1. As in 1+1d, the 2+1d dualities considered here change the quantum phases; for example, in the anisotropic limit the XXZ model with a unique gapped ground state is dual to a $\text{Rep}(S3)$ model with $D(S3)$ topological order in the same limit. Therefore, I generally expect duality implementations to require linear-depth unitary circuits. Some 1+1d dualities can be realized via constant-depth circuits supplemented with measurements (https://arxiv.org/abs/2311.01439), and extending this to 2+1d is an interesting direction. I’ve expanded the discussion of circuit realizations in the Conclusions. Regarding the second point, I agree that duality mappings in 2+1d should correspond to functors between module 2-categories.
  2. Indeed, should be able to build 2+1d fixed-point fusion surface models from the data of a fusion 2-category and the 2+1d analogue of a special symmetric Frobenius algebra. As discussed in Section 5.3 in Inamura & Ohmori, taking the fusion 2-category 2Vec$_G$ and a separable algebra therein, i.e. a $G$-graded multifusion 1-category, produces a $G$ symmetry-enriched Levin-Wen model. In the present setting, which starts from a braided fusion category, the same mechanism should yield fixed-point Hamiltonians from commutative special symmetric Frobenius algebras in the braided category (braiding is the minimal extra structure needed in 2+1d). As you note, interpolating between different algebras would offer a natural route to critical transitions between different topological phases.

---

## Editorial Decision

resubmitted